# Optimizing Inference-Time Reasoning in LLMs via Retrieval-Augmented Reflection

## Abstract

Empowering LLMs to improve their performance through increased inference-time computation is a crucial step in developing self-improving agents capable of operating in open-ended natural language contexts. In this paper, we explore how iteratively revising a chain of thoughts guided by information retrieval significantly improves large language models' reasoning ability in challenging tasks, while hugely mitigating hallucination. In particular, the proposed method — *retrieval-augmented reflection* (RaR) — revises the generation tokens step by step, leveraging multiple pieces of retrieved information relevant to the intermediate reasoning steps and the instruction. Applying RaR during inference-time to a various set of language models substantially improves their performances on various reasoning tasks; on relatively increasing scores by up to +16.4% on code generation, +11.6% on mathematical reasoning, and 29.1% on embodied task planning. Moreover, we find that with more inference-time computation given to the LLM for multi-times retrieval-augmented reflection, the LLM can continuously improve on various reasoning benchmarks. A small LM can surpass the performance of the LM with more than 10 times parameters, when giving more computation cost.

## 1 Introduction

Large Language Models (LLMs) have achieved fruitful progress on various natural language reasoning tasks (Wei et al., 2022; Yao et al., 2022; Wang et al., 2023a; Zhou et al., 2023; Brown et al., 2020), especially when combining large-scale models (Team, 2022; OpenAI, 2023) with sophisticated prompting strategies, notably chain-of-thought (CoT) prompting (Wei et al., 2022; Kojima et al., 2022). However, there have been increasing concerns about the factual correctness of LLMs reasoning, citing the possible hallucinations in model responses (Rawte et al., 2023) or the intermediate reasoning paths, *i.e.* CoTs (Dhuliawala et al., 2023). This issue becomes more significant when it comes to zero-shot CoT prompting, aka. "let's think step-by-step" (Kojima et al., 2022) and long-horizon generation tasks that require multi-step and context-aware reasoning, including code generation, task planning, mathematical reasoning, *etc*. Factually valid intermediate thoughts could be critical to the successful completion of these tasks.

Several prompting techniques have been proposed to mitigate this issue, one promising direction, Retrieval Augmented Generation (RAG) (Lewis et al., 2020b) seeks insights from human reasoning (Holyoak & Morrison, 2012), and utilizes retrieved information to facilitate more factually grounded reasoning. In this paper, we explore how to synergize RAG with

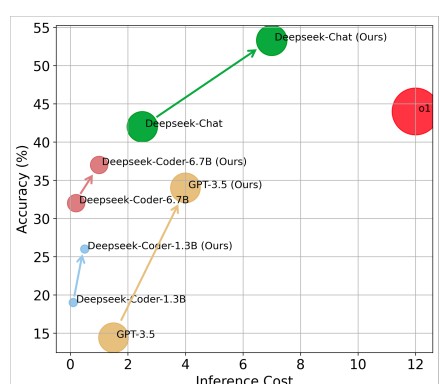

Figure 1: **Performance vs. Inference Cost for various models on the ClassEval benchmark (Du et al., 2023).** The size of each circle reflects the model parameters. The chart highlights the trade-off between inference-time computation and performance, demonstrating the effectiveness of different model architectures, with RaR-enhanced models generally achieving better performance for the same model scale. We also list the latest OpenAI O1 model (OpenAI, 2024), and our methods achieve better performance with less inference-time computation.

sophisticated long-horizon reasoning. Our intuition is that the hallucination within the intermediate reasoning process could be alleviated through the help of outside knowledge. The resulting prompting strategy, *retrieval-augmented reflection* (RaR), comprises two key ideas. Firstly, the initial zero-shot CoT produced by LLMs along with the original task prompt will be used as queries to retrieve relevant information that could help revise the possibly flawed CoT. Secondly, instead of retrieving and revising with the full CoT and producing the final response at once, we devise a progressive approach, where LLMs produce the response step-by-step following the CoT (a series of subtasks), and only the current thought step will be revised based on the information retrieved with task prompt, the current and the past CoTs. This strategy can be an analogy to the human reasoning process: we utilize outside knowledge to adjust our step-by-step thinking during complex long-horizon problem-solving (Holyoak & Morrison, 2012).

We observe that hallucinations in model outputs often originate from errors in earlier tokens, which can propagate and lead to incorrect final results. To tackle this issue, we extend retrieval-augmented generation to retrieval-augmented reflection. In RaR, the language model retrieves external information to verify and revise the original generation. This modification allows the model to correct errors in previously generated tokens—something traditional generation pipelines cannot achieve. By adopting retrieval-augmented reflection, we enable the model to iteratively improve its reasoning process in real-time, making it highly scalable with increased inference-time computation. This approach resolves the limitation of not being able to revise erroneous tokens generated earlier in the process, thus providing more reliable outputs without requiring model parameter adjustments (OpenAI, 2024). Different from methods required reinforcement learning to perform better reasoning during inference-time computation (OpenAI, 2024), our method does not require any modifications to the original model parameters; it only needs to provide more inference-time computation tokens, allowing the LM to automatically verify and revise the generation. As shown in Figure 1, RaR demonstrates better scalability performance.

We evaluate Retrieval-augmented Reflection (RaR) on a wide collection of challenging long-horizon tasks, including code generation, mathematical reasoning, embodied task planning, and creative writing. We employ several LLMs of varied scales: GPT-3.5 (Brown et al., 2020), GPT-4 (OpenAI, 2023), Deepseek-Coder (Zhu et al., 2024), Llama-3 (AI@Meta, 2024) and Gemma (Team et al., 2024). The results indicate that combing RaR with these LLMs elicits strong advantages over vanilla CoT prompting and RAG approaches. In particular, we observe new state-of-the-art level of performances across our selection of tasks when given the same maximum inference-time token limitation: 1) code generation: ClassEval (+16.4%), HumanEval (+4.7%), HumanEval+ (+2.4%), MBPP (+2.5%), MBPP+ (+4.2%); 2) mathematical reasoning problems: GSM8K (+11.6%), and GSMHard (+3.0%); 3) Minecraft task planning (+2.2% on accuracy); 4) QA (+16.44% on accuracy). Our scaling experiments show that RaR can be scalable with more inference-time computation and model parameters. By allowing for more inference-time tokens, RaR can achieve up to +33.3%, +12.2%, and +16.2% relative improvements in code generation, math reasoning, and task planning benchmarks, respectively. Our additional ablation studies further confirm the crucial roles played by the two key ingredients of RaR: revising CoT using RAG and progressive revision & generation. This work reveals how can LLMs revise their reasoning process in a zero-shot fashion with the help of outside knowledge, just as what humans do.

## 2 RELATED WORKS

**Retrieval-augmented Generation (RAG).** Recently, RAG has gained popularity for boosting the performance of LLMs by guiding their generation process using the retrieved knowledge (Zhao et al., 2023). Without updating model parameters that may be expensive (Lewis et al., 2020a) or unstable (Ke et al., 2022b;a), RAG is a cost-effective way for LLMs to interact with the external world (Gu et al., 2018; Lewis et al., 2020a). RAG is widely applied to downstream tasks, such as code generation (Zhou et al., 2022b; Lu et al., 2022; Nashid et al., 2023), question answering (Baek et al., 2023; Siriwardhana et al., 2023), and creative writing (Wen et al., 2023; Asai et al., 2023).

**Reasoning-enhanced RAG.** Some recent works also leverage reasoning to enhance the performance of RAG (Li et al., 2023b). For example, IRCoT (Trivedi et al., 2022) exploits CoT to generate better queries for retrieval, IRGR (Ribeiro et al., 2022) performs iteratively retrieval to search for suitable premises for multi-hop QA, GEEK (Liu et al., 2023a) can choose to query external knowledge or

perform a single logical reasoning step in long-horizon generation tasks, and ITRG (Feng et al., 2023a) performs retrieval based on the last-step generation. Active RAG (Jiang et al., 2023) also utilizes reasoning to enhance the quality of language model retrieval for better completion of QA tasks. These previous RAG methods simply adopt a single query to retrieve the knowledge for question-answering tasks (Gao et al., 2023; Feng et al., 2023b; Jiang et al., 2023; Yu et al., 2023). Our proposed RaR focuses on retrieval to refine LLM reasoning outputs and ensure consistent results. The reasoning and retrieval processes are interlinked to improve both aspects, as shown in Figure 2. Our approach is evaluated across long-horizon content generation tasks such as code generation, math reasoning, embodied planning, and creative writing.

**Language Model for Reasoning.** The advancement of reasoning in language models has seen notable methodologies emerge since CoT was proposed by Wei et al. (2022), which showcased LMs' ability to generate self-derived problem-solving strategies. This foundational work spurred further innovations such as the least-to-most prompting (Zhou et al., 2022a), zero-shot CoT (Kojima et al., 2022), self-consistency (Wang et al., 2022), zero-shot CoT without prompting (Wang & Zhou, 2024). Moving beyond basic prompting, Creswell et al. (2022) introduced the Selection-Inference framework, while Zelikman et al. (2022) developed STaR to refine reasoning through model finetuning. Creswell & Shanahan (2022) proposed a faithful reasoning model, segmenting reasoning into dedicated steps, similar to Scratchpad's approach by Nye et al. (2021) for enhancing multi-step computation. Tree-of-Thought (Yao et al., 2023) and Graph-of-Thought (Besta et al., 2023) also expand the reasoning paths into a complex structure instead of linear CoT. These methods usually aim to improve the reasoning ability of LLM by designing prompts or providing feedback from the environment to assist in better planning and decision-making (Wang et al., 2023c; Yao et al., 2022; Shinn et al., 2023; Li et al., 2023a; Zhang et al., 2023). However, RaR takes a different approach by using RAG to access external knowledge that can help LLM with its reasoning process.

# 3 RETRIEVAL AUGMENTED REFLECTION

Our goal is to support reasoning and generation while mitigating hallucination when using LLMs. To have satisfying performance on long-horizon tasks, two ingredients are indispensable. Firstly, access to factual information can be facilitated by retrieval. Secondly, appropriate intermediate steps that outline a scratchpad to finish complex tasks, can be facilitated by CoT. Yet, a naive combination of the two would not necessarily yield improvements. Three questions still persist: (1) what is relevant information to retrieve; (2) how to effectively correct reasoning steps with relevant factual information; (3) can this combination be scalable with more inference-time computation. To better appreciate our method and why our method can address these questions, we first provide a brief introduction of basic RaR and interactive RaR.

## 3.1 RETRIEVAL AUGMENTED REFLECTION

We first describe the pipeline of the **Retrieval-augmented Reflection** (RaR) method. The key steps of the RaR process are as follows:

First, given an input question or instruction $x$, we first generate an initial coarse response $y^{\text{raw}} \sim p_{\text{LM}}(\cdot \mid x)$ using the language model $p_{\text{LM}}$, where $p_{\text{LM}}$ denotes the probabilistic output of the language model. To enhance the quality of initial response $y_{\text{raw}}$, we construct a retrieval query $q \sim p_{\text{LM}}(\cdot \mid x, y^{\text{raw}})$ by leveraging both the input question $x$ and the initial coarse response $y^{\text{raw}}$. We then retrieve the relevant document $V^k$ from a set of $K$ candidate documents $V = \{V^1, V^2, \dots, V^K\}$. Generally, the retrieval process involves selecting the document that maximizes a similarity function $\text{sim}(q, V^k)$, defined over the query embedding $\text{emb}(q) \in \mathbb{R}^d$ and the documents embeddings $\text{emb}(V^k) \in \mathbb{R}^d$:

$$V^k = \arg\max_{k \in \{1,\dots,K\}} \text{sim}(q, V^k) = \arg\max_{k \in \{1,\dots,K\}} \frac{\text{emb}(q) \cdot \text{emb}(V^k)}{\|\text{emb}(q)\|\|\text{emb}(V^k)\|}, \tag{1}$$

where $d$ is the dimension of text embeddings (Reimers & Gurevych, 2019). Finally, the retrieved document $V^k$ is then used to prompt the language model to reflect the initial response $y^{\text{raw}}$ and output reflection $y^{\text{reflection}} \sim p_{\text{LM}}(\cdot \mid x, y^{\text{raw}}, V^k)$. The reflection process identifies potential errors or hallucinations in $y^{\text{raw}}$. Based on the reflection output $y^{\text{reflection}}$, the language model then refine the

---

**Algorithm 1** Iterative Retrieval-augmented Reflection (RaR)

---

**Input:** Task Prompt $x$, Autoregressive Large Language Model $p_\theta$, Number of Reasoning Steps $n$

1: $y^{\text{raw}}, \{y_i^{\text{thought}}\}_{i=1}^{i=n} \leftarrow p_\theta(\cdot|I)$ ▷ Generate initial response $y^{\text{raw}}$ and step-by-step reasoning $y^{\text{thought}}$
2: $y_0^{\text{RaR}} \leftarrow$ None ▷ Initialize the intermediate RaR response
3: **for** $i = 1$ to $n$ **do**
4: $\quad q_i^{\text{inter}} \leftarrow p_{\text{query}}(x, y_{i-1}^{\text{RaR}}, y_i^{\text{thought}})$ ▷ Generate query for intermediate reasoning step $i$
5: $\quad V_i^{\text{inter}} \leftarrow$ RetrieveFromCorpus$(q_i^{\text{inter}})$ ▷ Retrieve related document for step $i$
6: $\quad y_i^{\text{RaR}} \leftarrow p_\theta(\cdot|I, T_{\leq i-1}^{\text{thought}}, K_i^{\text{inter}})$ ▷ Reflect and refine step $i$
7: **end for**
8: $y_{n+1}^{\text{RaR}} \leftarrow p_\theta(\cdot|I, y^{\text{RaR}}, T^{\text{raw}}, K_{n+1}^{\text{iter}})$ ▷ Initialize the overall RaR response
9: $j \leftarrow n + 1$ ▷ Start Overall Response RaR
10: **repeat**
11: $\quad q_j^{\text{iter}} \leftarrow p_{\text{query}}(x, y_j^{\text{RaR}})$ ▷ Generate query for full reflection step $j$
12: $\quad V_j^{\text{iter}} \leftarrow$ RetrieveFromCorpus$(q_j^{\text{iter}})$ ▷ Retrieve additional related documents
13: $\quad y_j^{\text{RaR}} \leftarrow p_\theta(\cdot|x, y_{j-1}^{\text{RaR}}, V_j^{\text{iter}})$ ▷ Iteratively refine full response
14: $\quad j \leftarrow j + 1$
15: **until** Convergence (e.g., $m$ identical generations) or maximum token limit reached
16: **return** $T^\star$ ▷ Output refined response $y^{\text{RaR}}$ as the final generation

---

initial raw coarse response, incorporating the corrective information from $V^k$:

$$\text{RAG}: \quad y^{\text{RAG}} \sim p_{\text{LM}}(\cdot \mid x, V^k), \ V^k = \arg \max_{k \in \{1,...,K\}} \text{sim}(x, V^k),$$

$$\text{RaR}: \quad y^{\text{RaR}} \sim p_{\text{LM}}(\cdot \mid x, y^{\text{raw}}, V^k, y^{\text{reflection}}). \tag{2}$$

This pipeline synergizes the Retrieval-augmented Generation and self-reflection based on the same Language Model, which is different from RAG methods (Lewis et al., 2020b). To improve the performance, we further increase the number of RaR iterations through iterative integration of retrieval-augmented generation and self-reflective reasoning.

## 3.2 ITERACTIVE RETRIEVAL AUGMENTED REFLECTION

To further scale up reasoning accuracy and overall response quality, we extend the basic RaR framework to an iterative version, referred to as **Iterative RaR**. This approach enables the model to repeatedly retrieve and reflect on intermediate reasoning steps and the overall response in multiple iterations, progressively improving its performance. The workflow of Iterative RaR comprises two key phases:

**Reflection on Intermediate Reasoning Steps.** In this phase, RaR focuses on improving the step-by-step reasoning process by reflecting on intermediate steps. We first enhance the instruction $x$ with a CoT prompt to form an augmented instruction $x^*$. The language model then generates a reasoning process and the raw response, represented as:

$$(y^{\text{thought}}, y^{\text{raw}}) \sim p_{\text{LM}}(\cdot \mid x^*), \tag{3}$$

where $y^{\text{thought}} = \{y_1^{\text{thought}}, \ldots, y_J^{\text{thought}}\}$ denotes the step-by-step reasoning process, divided into $J$ sections corresponding to $J$ reasoning steps. For each iteration $i$, the retrieval query $q^i$ is constructed based on the causal history of reasoning steps up to step $i$, while masking future steps $j > i$. The query generation is expressed as:

$$q^i \sim p_{\text{LM}}(\cdot \mid x^*, \{y_j^{\text{thought}}\}_{j=1}^{j<=i}), i = 1, \ldots, J. \tag{4}$$

To improve efficiency, this retrieval based on intermediate reasoning steps is parallelized through causal mask (Vaswani et al., 2017), enabling the generation of queries for different reasoning steps simultaneously. And the retrieved documents $\{V_k^j\}_{j=1}^J$ are used to review and correct any potential errors in the reasoning steps.

$$y_i^{\text{RaR}} \sim \begin{cases} p_{\text{LM}}(\cdot \mid x, y_i^{\text{thought}}, V_i^k, y_i^{\text{reflection}}), & \text{if } i = 1, \\ p_{\text{LM}}(\cdot \mid x, y_{i-1}^{\text{RaR}}, y_i^{\text{thought}}, V_i^k, y_i^{\text{reflection}}), & \text{if } 1 < i < J, \\ p_{\text{LM}}(\cdot \mid x, y_{i-1}^{\text{RaR}}, y_i^{\text{thought}}, y^{\text{raw}}, V_i^k, y_i^{\text{reflection}}), & \text{if } i = J. \end{cases} \tag{5}$$

This allows for correcting errors in the original thoughts $y^{\text{thought}}$ by continually consulting different reference texts and ensuring that the most accurate and relevant information informs each step of reasoning. Previous methods have demonstrated that in tasks involving long-term planning and rigorous reasoning, like mathematical reasoning (Lightman et al., 2023) and embodied planning (Yao et al., 2022; Shinn et al., 2023), supervision of intermediate processes is necessary to ensure the accuracy of model outputs. However, these approaches typically rely on feedback from humans or the environment, which can be costly in situations where exploration and annotation expenses are high, such as in safe decision-making scenarios (Gu et al., 2022). In contrast, RaR can automatically access relevant information from external sources to validate and revise the content of model outputs through a retrieval process. This allows RaR to autonomously verify each step without requiring human labels (Lightman et al., 2023), which explains its significant success in mathematical reasoning.

**Refinement of Overall Response.** After refining the intermediate reasoning steps, RaR observes the full response $y_J^{\text{RaR}}$ with refined intermediate steps, obtained from the corrected reasoning process. The full response is then used to produce query $q_i \sim p_{\text{LM}}(\cdot \mid x^*, y_J^{\text{RaR}})$ about overall structure to identify potential errors or inconsistencies in the overall structure, where $i > J$. The retrieved documents guide the refinement of the complete response, ensuring consistency and correctness across the entire output. The final response will be formulated as:

$$y_i^{\text{RaR}} \sim p_{\text{LM}}(\cdot \mid x, y_{i-1}^{\text{RaR}}, V_k^i, y_i^{\text{reflection}}), i > J. \tag{6}$$

We can repeat the RaR more times for better refinement of final response with more inference-time computation. The iterative process continues until the response reaches the maximum token limitation or the reflection process produces $m$ consecutive identical generations, indicating convergence. In practice, $m$ is set as 3.

Our hypothesis why our method can address the two problems mentioned at the beginning of this section is as follows. Firstly, the most straightforward way to know what information will be used in complex reasoning is to "see" the reasoning steps. Our approach leverages all the generated thoughts along with the task prompt to provide more clues for more effective retrieval. Secondly, some information cannot be directly retrieved, especially information related to the final answer to a hard complex question. Instead, retrieval of information relevant to intermediate questions, which are assumed to be easier, is more accessible. Thanks to the compositional nature of many reasoning tasks, an iterative retrieval process could also be more effective. Thirdly, correcting potential hallucinations needs to be targeted. Revising a complete CoT with RAG could introduce errors at otherwise already-correct steps. Revising every step one by one could be more reliable. The first two points address question (1) and the last point addresses question (2). Quantitative evidence can be found in our ablation studies in Section 4.4.

## 3.3 OPTIMIZING RaR BY SCALING INFERENCE-TIME COMPUTATION

In this section, we explore how to improve the reasoning performance of LLMs by scaling inference-time computation. While the native approach to scaling inference-time performance typically involves increasing the parameters of the LLM itself, we present an alternative method using Iterative Retrieval-augmented Reflection (RaR) that scales computation by augmenting input and output tokens iteratively.

**Native Scaling via Model Size.** The conventional approach to improving LLM performance, referred to as DIRECT Scaling, relies on increasing the number of model parameters, as suggested by the LLM scaling laws (Kaplan et al., 2020). Larger models generally exhibit better performance with higher computational costs during inference. However, this method comes with significant drawbacks, such as the increased burden of training large-scale models and the accompanying demand for vast amounts of data (Kaplan et al., 2020).

**Inference-time Scaling without Increasing Model Parameters.** Recent research shows that inference-time performance can be improved for LLMs with fixed parameters by increasing the computational cost through additional *input tokens* or *output tokens* during inference (Snell et al., 2024). **Scaling input tokens** involves extending the user prompt with more tokens. Few-shot CoT adds more demonstrations (instruction-response pair) to the prompt, e.g., increasing from 1-shot to 5-shot CoT, enables better contextual reasoning for complex tasks (Wei et al., 2022). RAG retrieves and adds more related documents to the user prompt. For instance, increasing the top-$k$

documents from 1-shot RAG to 5-shot RAG can provide richer contextual information for reasoning. **Scaling output tokens** involves generating more detailed outputs or sampling more generations. For example, Self-Consistency will generate multiple outputs in parallel and merge them into a final response (Wang et al., 2023a).

Iterative RaR simultaneously scales **both input and output tokens** by performing multiple rounds of retrieval and reflective reasoning. Specifically, for $n$ iterations, RaR adds $n \cdot$ top-$k$ related documents to the input tokens and generates more than $n$-times the output tokens during reasoning. This iterative process enables a gradual refinement of the reasoning process and response quality. As shown in our experiments, the performance of RaR improves consistently with increasing computational cost.

Scaling input and output tokens at inference time often encounters the long-context problem, wherein the fixed context length of LLMs restricts the number of tokens that can be processed (Brown et al., 2020). When the token count approaches the model's maximum limit, performance degradation can occur (Li et al., 2024). RaR mitigates this issue by iteratively performing retrieval and reflection, thereby avoiding a single, overly lengthy context and maintaining high performance.

It is worth noting that some approaches employ agent systems that utilize multiple prompts within a single LLM to create complex pipelines for answering questions (Yao et al., 2022; Gravitas, 2024). However, such methods are compositions of foundational techniques and are not directly comparable to standalone reasoning and generation methods like RaR. For this reason, we exclude agent-based approaches from our discussions and experiments.

## 4    EXPERIMENTS

We evaluate our proposed method RaR on a variety of benchmarks that emphasize LLM generation and reasoning. Previous methods have struggled with these benchmarks, often producing "hallucinated" steps in LLM outputs that do not align with the original query or are clearly incorrect. For a detailed discussion, please refer to subsection 4.3 (case analysis). Due to space limitations, we do not present each benchmark setting or discuss our results extensively for each benchmark. Instead, this section offers a thorough showcase of our method's performance, shedding light on the preliminary empirical analysis of when our method succeeds and when it falls short.

### 4.1    EXPERIMENTAL SETUPS

**Benchmarks.** We adopt four groups of benchmarks including class-level code generation, math reasoning, task planning, and question answering.[1] **Code Generation** benchmarks include ClassEval (Du et al., 2023), HumanEval (Chen et al., 2021), HumanEval+ (Liu et al., 2023b), MBPP (Austin et al., 2021), and MBPP+ (Liu et al., 2023b). These benchmarks encompass a wide range of programming problems, from simple function implementations to more complex class-level generation challenges, providing a robust testbed for assessing generative and reasoning capabilities. **Mathematical Reasoning** evaluation is conducted on GSM8K and GSM-HARD dataset, which comprises thousands of multi-step mathematical problems (Cobbe et al., 2021; Gao et al., 2022). **Task Planning** are evaluated on open-ended environments Minecraft. A set of 100+ tasks ranging from simple objectives to challenging diamond objectives are evaluated through MC-TextWorld (Lin et al., 2023). **Question-answering (QA)** tasks are assessed using TriviaQA (Joshi et al., 2017) benchmark, where systems are required to answer various questions related to factual knowledge.

**Evaluation Metrics.** For fair evaluation, we report the best results achieved by all methods under the given maximum token limitation in Table 1, and the performance of different methods under different inference computation costs is reported in Figure 2. For code generation benchmarks, we use classical pass rate pass@5 as the evaluation metrics (Chen et al., 2021; Liu et al., 2023b). In ClassEval, we tested the generation pass rates for class-level and method-level (function) separately. We compute accuracy to evaluate every question in mathematical reasoning tasks, aligning with the established

---

[1] We used `bigcode-evaluation` as the tool library for code evaluation. The pass@1 result of DIRECT in the table is slightly different from the result in the bigcode leaderboard, because we tested our pass@1 five times in our original setup and calculated the average value. We used the same settings as DIRECT in all experiments and reported on the relative improvement of RaR compared to baselines to promise fair evaluation and comparison.

Table 1: The evaluation results cover various benchmarks such as code generation, math reasoning, and task planning. All evaluations were conducted using the GPT-3.5-turbo model from the OpenAI API, which has a maximum token length of 4096.

| Token Limitation | Method | ClassEval Function | ClassEval Class | HumanEval | HumanEval+ | MBPP | MBPP+ | GSM8K | GSMHard | Planning |
|---|---|---|---|---|---|---|---|---|---|---|
| | DIRECT (gpt-3.5) | 14.4 | 23.8 | 72.5 | 70.5 | 72.9 | 64.0 | 65.9 | 51.3 | 19.3 |
| | **Input Scaling** | | | | | | | | | |
| | RAG (n-shot) | 24.3 | 53.1 | 76.2 | 70.5 | 70.5 | 68.0 | 61.8 | 56.8 | 33.0 |
| | IRCoT | 29.2 | 57.6 | 77.4 | - | - | - | - | 60.3 | 57.3 |
| | Active RAG | 25.7 | 55.3 | 75.8 | - | - | - | - | 61.3 | 59.4 |
| | **Output Scaling** | | | | | | | | | |
| 4K | Few-shot CoT | 16.0 | 36.2 | 75.8 | 74.8 | 65.4 | 62.9 | 63.8 | 44.7 | 49.3 |
| | Self-Refine | 21.4 | 46.4 | 75.8 | 74.2 | 69.4 | 65.6 | 65.8 | 55.4 | 50.5 |
| | Self-Consistency | 19.4 | 40.4 | 75.8 | 72.9 | 73.8 | 69.7 | 65.0 | 52.8 | 50.2 |
| | **Both Scaling** | | | | | | | | | |
| | RAG+CoT | 24.7 | 55.7 | 76.8 | 69.3 | - | - | 67.5 | 67.3 | 50.4 |
| | RaR (ours) | **34.0** | **66.9** | **80.4** | **76.0** | **74.7** | **72.6** | **75.3** | **69.3** | **60.5** |
| 8K | Self-Refine | 22.8 | 45.4 | 73.2 | 72.6 | 67.3 | 63.5 | 62.1 | 50.3 | 55.5 |
| | RaR (ours) | 37.3 | 66.9 | 81.3 | 79.4 | 76.9 | 74.9 | 75.8 | 69.3 | 76.7 |
| 16K | RaR (ours) | 38.9 | 67.2 | 82.5 | 79.6 | - | - | - | - | 76.7 |

metric for the GSM8K (Cobbe et al., 2021). For embodied planning tasks, we compute the plan execution success rate in MC-TextWorld as accuracy (Lin et al., 2023). For the QA benchmarks, we use the official accuracy metric (Asai et al., 2023) to evaluate all methods. These indicators are better the higher they are.

**Baselines.** To establish a comprehensive and equitable comparison landscape, we incorporate a suite of baseline methods. Our baselines, in addition to directly using LLM for sampling (DIRECT), are divided into three groups: Scaling input tokens, which includes RAG (Lewis et al., 2020b), IRCoT (Trivedi et al., 2022), Active-RAG (Jiang et al., 2023); scaling output tokens, which includes CoT (Wei et al., 2022), Self-consistency (Wang et al., 2023a), Self-refine (Madaan et al., 2024); and scaling both, including RAG+CoT and RaR. For specific scaling methods of each approach, please refer to Section 3.3. For each method, we will scale them to stay within the permitted maximum token limit, e,g, by adjusting the sample size $n$ for self-consistency and the number of documents $k$ in RAG. For the QA benchmark, we also list the result from Self-RAG (Asai et al., 2023) in Figure 2 (a). For different methods, the same language model is used as the base model. All methods in the Table 1 are evaluated with gpt-3.5-turbo. To ensure a fair comparison, none of the methods used examples from the benchmark as demonstrations for in-context learning.

**Model and RAG Settings.** RaR leverages the capabilities of Retrieval-Augmented Generation methods, which enhance the performance of language models by integrating external knowledge sources. Specifically, we employed the `codeparrot/github-jupyter` dataset as our primary search vector library for code generation and mathematical reasoning tasks. For embodied planning tasks in Minecraft, we utilized the Minecraft Wiki[2] and DigMinecraft[3] websites as the information sources accessible to the LLMs. For QA benchmarks, we use the wiki pages as the retrieval library, which is consistent with Asai et al. (2023). We utilized OpenAI's `text-embedding-ada-002` API service for all embedding calculations across different methods and base models.

Acknowledging the risk of benchmark contamination (an issue where the code library may contain solutions to the exact problems being evaluated), we adopted a rigorous pre-processing methodology as described by Guo et al. (2024). The potential implications of benchmark contamination, along with the effectiveness of our pre-processing strategy, are discussed in detail in Appendix D.

## 4.2 RESULTS

The code generation, math reasoning and task planning results are presented in Table 1 and QA results are presented in Figure 2 (a), which demonstrate the comprehensive evaluation of the RaR across multiple benchmarks. RaR consistently outperforms the other methods across the majority of

---

[2] https://minecraft.wiki/
[3] https://www.digminecraft.com/

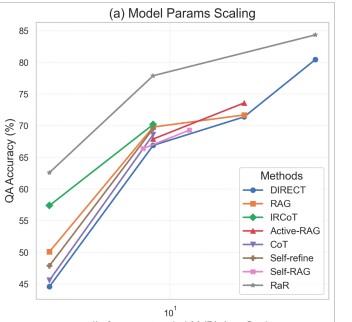 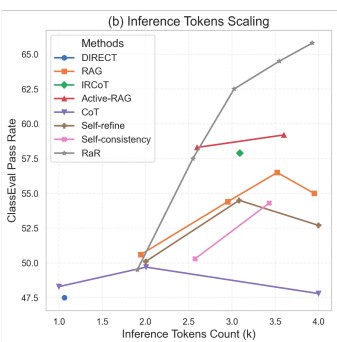 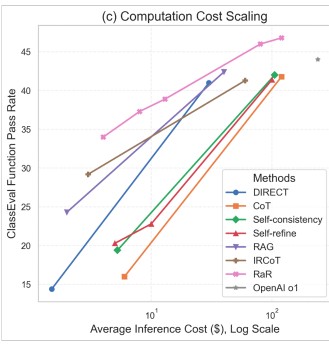

Figure 2: Evaluation results on different methods with (a) scaling model parameters, (b) scaling inference tokens, and (c) average computation cost. The language model in (b) is Deepseek-Coder 6.7B (Zhu et al., 2024). The language models in (c) are all OpenAI models including gpt-3.5 (Brown et al., 2020), gpt-4 (OpenAI, 2023), and openai o1 (OpenAI, 2024). The computation cost in (c) is computed with OpenAI Pricing.

the benchmarks and metrics, showcasing its superior ability in reasoning. For the most challenging class-level generation tasks in ClassEval, RaR based on ChatGPT outperforms the base model with more than 1.8 times improvements. For the method-level generation tasks in the HumanEval and HumanEval+ benchmarks of code generation, RaR achieves remarkable improvements in pass@5 rates, indicating a significant enhancement in first-attempt accuracy and within the top five attempts. For example, on the HumanEval benchmark, RaR improves pass@5 by up to 25.68% relative to the base models' performances. This trend is observed across different underlying base models, highlighting RaR's effectiveness regardless of the initial model's capabilities. For mathematical reasoning tasks, RaR demonstrates a significant relative improvement, with a 14.44% increase in accuracy on GSM8K and a remarkable 35.27% relative increase on GSMHard, culminating in an overall average improvement of 23% when deployed on the GPT-3.5 model. RaR significantly outperforms all other methods on task planning tasks in Minecraft, achieving the highest scores with 76.67±8.02% for accuracy and 88.73% for partial accuracy, demonstrating its superior ability to generate feasible and contextually appropriate plans in the complex open-world environment. RaR has achieved state-of-the-art results on the QA benchmark. Specifically, the RaR method, based on the 8B model, outperforms the larger 27B model in terms of direct output results despite having fewer parameters as shown in Figure 2 (a). While other methods like self-RAG (Asai et al., 2023) and Active-RAG (Jiang et al., 2023) also incorporate reasoning methods and RAG during inference, they fall short in performance compared to RaR. This highlights the effectiveness of the RaR method in leveraging retrieved content for reflective reasoning.

The tasks are extremely diverse, while RaR can have consistent improvements over all baselines. These results underline the advantages of RaR's approach, which leverages iterative refinement of retrieval queries based on evolving reasoning thoughts. This strategy not only enhances the relevance and quality of the information retrieved but also significantly improves the accuracy and efficiency of the generated context.

### 4.3 SCALING EXPERIMENTS

We explored the performance scaling of the RaR technique from three key perspectives: (1) increasing the base model parameters, (2) increasing the inference-time tokens, and (3) increasing the inference-time computation cost (influenced by both token number and model parameters) which correspond to more iterations of Iterative Retrieval-Augmented Reflection during inference. These experiments were conducted on the Trivial QA Benchmark (Joshi et al., 2017) and the ClassEval Code Generation Benchmark (Du et al., 2023). The results are demonstrated in Figure 2.

**Scaling Model Parameters.** In the QA benchmark, we utilized language models with varying parameter sizes—Gemma-2-2B, Llama-3-8B, Gemma-2-27B, and Llama-3-70B—as base models (Team et al., 2024; AI@Meta, 2024). Our findings demonstrate that RaR consistently enhances performance as the model scale increases. This trend was observed across all base model sizes, with RaR maintaining a significant performance advantage over both the Gemma and Llama models in the QA task. Under the same model parameters, RaR always performs the best. Importantly, the

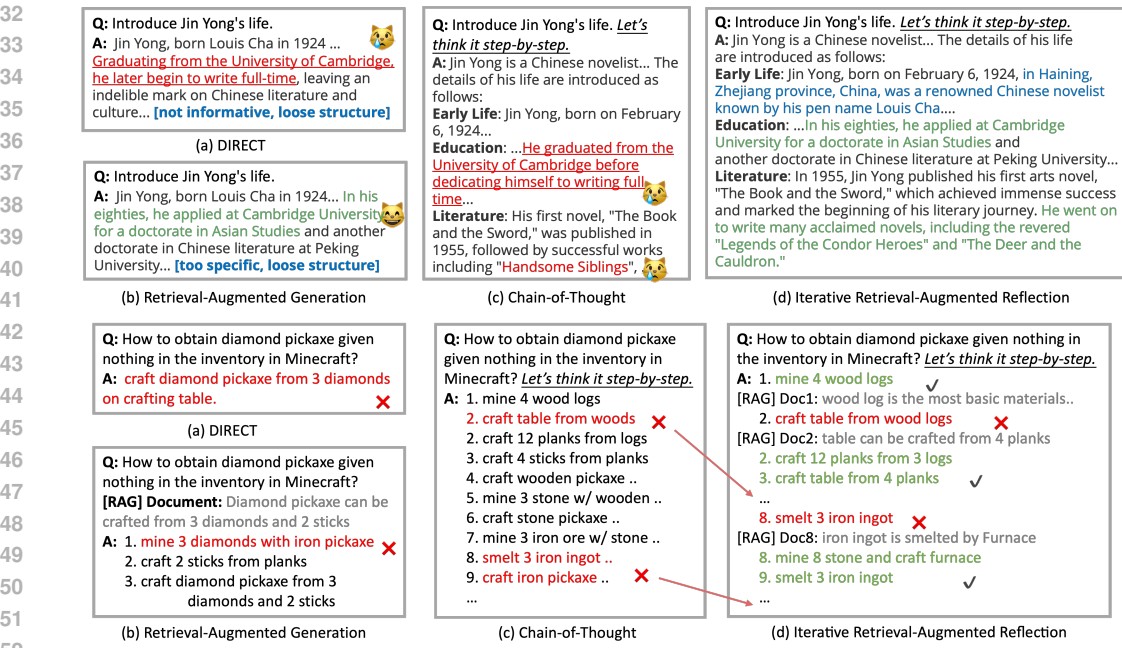

Figure 3: Case analysis on long-horizon task planning and question answering. RaR improves upon CoT's initial answers by continuously refining thoughts with intermediate step retrieval and overall retrieval, aligning closely with task progression and relevant item knowledge.

more reasoning-intensive the task, the more pronounced the performance gains achieved by RaR, underscoring its ability to improve reasoning capabilities, which are typically more pronounced in larger models.

**Scaling Inference Tokens.** Additionally, we investigated the impact of stricter inference-time tokens on the performance of different methods. We utilize the DeepSeek-Coder-6.7B (Zhu et al., 2024) on ClassEval method-level pass rates (Du et al., 2023) to analyze all methods and present the performance results along with the corresponding tokens in Figure 2 (b). Almost all methods show growth with the increase in inference-time tokens used. RaR performs worse than methods like IRCoT when the number of tokens used is less than 2k, because RaR has not yet completed modifications to all intermediate steps. However, when given more tokens up to 4k, RaR shows significant growth compared to other methods. Additionally, we found that methods like self-consistency and RAG experienced a decline in performance during the later stages of token growth. We speculate that this is due to these methods requiring a large number of tokens at once, leading to long text issues that cause performance degradation. In contrast, RaR, by using an iterative updating approach, does not encounter long text problems, thus showing continuous performance growth when given more tokens. This demonstrates that RaR has good scaling potential.

**Scaling Computation Cost.** Finally, we will also combine the model scale and the number of tokens used to examine the performance of different methods under computation cost scaling. The horizontal axis in Figure 2 (c) represents the average API price used, where all models utilize OpenAI models, including gpt-3.5-turbo, gpt-4, and openai-o1. All pricing is based on OpenAI's official pricing[4], calculated according to the actual consumption of prompt tokens and generation tokens. We found that RaR is the most economical choice, as it is nearly the best under the same computation cost. Additionally, we discovered that with less computation cost, the RaR based on gpt-4 demonstrated performance exceeding that of openai o1.

### 4.4 ABLATION STUDY

**Ablation on retrieval in RaR.** In this ablation study, we investigate the influence of various retrieval strategies on the efficacy of RaR, focusing on the optimization of content retrieval for

---

[4]https://openai.com/api/pricing/

Table 2: Comparative Impact of Retrieval Strategies on RaR Performance.

| Method | HumanEval | | HumanEval+ | |
|---|---|---|---|---|
| | pass@1($\Delta$) ↑ | pass@5($\Delta$) ↑ | pass@1($\Delta$) ↑ | pass@5($\Delta$) ↑ |
| Baseline | 50.6% | 76.2% | 48.2% | 70.5% |
| CoT+RAG | 53.9(+3.3)% | 76.8(+0.6)% | 51.3(+3.1)% | 69.3(-1.2)% |
| RaR | 59.2(+8.7)% | 80.4(+7.9)% | 56.3(+8.2)% | 76.0(+5.5)% |

Table 3: Ablation Study on Causal vs. Non-Causal Reasoning in RaR.

| Method | HumanEval | | HumanEval+ | |
|---|---|---|---|---|
| | pass@1($\Delta$) ↑ | pass@5($\Delta$) ↑ | pass@1($\Delta$) ↑ | pass@5($\Delta$) ↑ |
| Baseline | 47.3% | 75.8% | 41.7% | 74.8% |
| Non-Causal | 57.3(+10.0)% | 78.0(+2.1)% | 54.9(+13.2)% | 74.8(+0.0)% |
| Causal | 59.2(+11.9)% | 80.4(+4.6)% | 56.3(+14.6)% | 76.0(+1.2)% |

improving generative outputs. The experimental results, detailed in Table 2, highlight the significant advancements achieved through the iterative refinement of retrieval queries in RaR compared to baseline methods. The baseline denoted as RAG-1, employs a direct approach by using the question itself as the retrieval query. In contrast, CoT+RAG enhances this process by utilizing the entirety of the reasoning thoughts output by the language model as the query, aiming for a broader contextual understanding. However, RaR introduces a more dynamic method by employing continuously modified parts of reasoning thoughts as queries, which allows for a more focused and relevant information retrieval process. The comparative analysis shows that RaR surpasses both the baseline and the CoT+RAG method in terms of pass@1 and pass@5 metrics across the HumanEval and HumanEval+ benchmarks. Specifically, RaR demonstrates an 8.7 percentage point increase in pass@1 and a 7.9 percentage point increase in pass@5 over the baseline in the HumanEval benchmark, and similarly impressive gains in the HumanEval+ benchmark. These improvements underscore the effectiveness of RaR's retrieval strategy, which by iteratively refining next queries based on evolving reasoning thoughts and previous queries, ensures the retrieval of highly pertinent information. This process not only enhances the relevance of the information retrieved but also significantly improves the quality and accuracy of the final generated outputs. The results firmly establish the superiority of RaR's dynamic retrieval method in leveraging contextual nuances to drive more precise and effective generative processes.

**Ablation on causal reasoning in RaR.** In this ablation study, we systematically examine the impact of causal and non-causal reasoning approaches on the performance of the RaR system, with the Chain of Thought (CoT) serving as our baseline. Our findings, as summarized in Table 3, reveal significant enhancements in generation capabilities when incorporating causal reasoning techniques. Specifically, the causal approach, which iteratively performs reasoning and retrieval, leads to notable improvements in both pass@1 and pass@5 metrics across HumanEval and HumanEval+ benchmarks. For instance, the causal method outperforms the baseline (CoT) by 11.9 percentage points in pass@1 and by 4.6 percentage points in pass@5 on the HumanEval dataset. This approach contrasts with the non-causal method, which, although also surpassing the baseline, leverages the initial reasoning thought to directly retrieve all necessary steps and generate the final answer. The causal method's superior performance underscores the value of sequential reasoning and information retrieval in enhancing the accuracy and reliability of generated outputs. This iterative process likely aids in refining the search and reasoning steps based on continuously updated context, allowing for more precise and relevant information retrieval, which in turn supports more accurate final answers. These results firmly establish the efficacy of causal reasoning in long-horizon problem-solving tasks.

## 5 DISCUSSION ON LIMITATIONS AND CONCLUSION

One limitation of this work is that the performance of RaR relies on the chain-of-thought reasoning and in-context learning (or RAG) capability of the base LLM. Since this work does not involve any model training, the capability of base LLM will not change when applying RaR. Despite RaR achieves significant improvement on powerful LLMs such as GPT-3.5 and GPT-4, the effect on smaller and weaker LLMs is questionable. Another limitation of this work is that the performance of RaR also relies on the quality of the retrieved knowledge. Another limitation of this work is that the performance of RaR also relies on the quality of the retrieved knowledge.

We have presented Retrieval Augmented Reflection (RAR), a simple yet effective prompting strategy that synergies chain of thought (CoT) prompting and retrieval augmented generation (RAG) to address the challenging long-horizon reasoning and generation tasks. Our key ideas involve revising the zero-shot chain of thoughts produced by LLMs through RAG with the thoughts as queries, and causally revising the thoughts & generating the response progressively. RaR, a **zero-shot** prompting approach, has demonstrated significant advantages over vanilla CoT prompting, RAG, and other

baselines on challenging code generation, mathematics reasoning, embodied task planning, and creative writing tasks.

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

## A  TASK DETAILS

### A.1  CODE GENERATION

**Benchmarks.**  We select HumanEval (Chen et al., 2021), HumanEval+ (Liu et al., 2023b), MBPP (Austin et al., 2021), and MBPP+ (Liu et al., 2023b) as the code generation evaluation benchmark.  These benchmarks are commonly used to test the performance of code generation models, which are briefly introduced below:

- **HumanEval** consists of 164 Python programming problems, each with a function signature, docstring, body, and multiple unit tests (Chen et al., 2021).
- **HumanEval+** includes the same programming problems as HumanEval, but with an additional 80 times more unit tests for each of the 164 problems (Liu et al., 2023b).
- **MBPP** is a collection of approximately 1,000 Python programming problems that are intended to be solvable by beginner programmers. Each problem includes an English task description, a code solution, and three automated test cases. We assess the sample test set from index 11 to 175 (Austin et al., 2021).
- **MBPP+** consists of 399 tasks (Liu et al., 2023b), which are a subset of the original MBPP dataset.  Additionally, MBPP+ includes extra unit tests for each of the 399 problems (35 times more than the original MBPP). We utilized the first 164 questions as our test set.

These benchmarks encompass a wide range of programming problems, from simple function implementations to more complex algorithmic challenges, providing a robust testbed for assessing the generative capabilities of various models.

**Metrics.**  We adopt the `pass@k` metric for evaluating the efficacy of various code generation algorithms, following the methodology proposed by Chen et al. (2021) and extended by Liu et al. (2023b). This metric quantifies the rate at which generated code snippets successfully execute and pass all test cases, where $k$ represents the number of attempts or samples generated by the model for each problem.  This approach allows us to rigorously assess the precision and reliability of code generation models in producing functionally correct code across a diverse set of programming challenges.

**Baselines.**  To establish a comprehensive and equitable comparison landscape, we incorporate a suite of baseline methods and diverse code generation models. Our baselines include the original code generation language models, referred to as DIRECT, and the Retrieval-Augmented Generation (RAG) methodology with $n$ retrieved examples, instantiated in both single-shot (1 shot) and multi-shot (5 shots) configurations, as documented by Lewis et al. (2020b). Additionally, we examine the zero-shot CoT (CoT) approach, as conceptualized by Kojima et al. (2022), which simulates a step-by-step reasoning process to facilitate complex problem-solving tasks under zero demonstration. To ensure a fair comparison, none of the methods used examples from the benchmark as demonstrations for in-context learning.

The diversity of our evaluation is further enriched by testing across various language models with differing capacities, including CodeLlama-7b (Rozière et al., 2023), along with Chat-GPT(`gpt-3.5-turbo`) (Ouyang et al., 2022), and the more advanced GPT-4(`gpt-4`) model (OpenAI, 2023). Recognizing the potential format discrepancies in code outputs, especially considering that models like `gpt-3.5-turbo` and `gpt-4` may produce code in markdown format which is not immediately executable, we implement post-processing steps to convert the original language model outputs into a form that can be executed within a sandbox environment. This normalization ensures that all models are evaluated under uniform execution conditions, thereby facilitating a fair and direct comparison of their code generation capabilities. Through this methodological framework, we aim to provide a detailed and nuanced understanding of the performance landscape across a spectrum of LLM-driven code generation approaches.

**RAG Settings.** RaR leverages the capabilities of Retrieval-Augmented Generation methods, which enhance the performance of language models by integrating external knowledge sources. Specifically, we employed the `codeparrot/github-jupyter` dataset as our primary search vector library. This dataset is a comprehensive compilation of 452k markdown and code pairs, meticulously extracted from Jupyter notebooks hosted on GitHub BigQuery, representing a rich repository of programming

knowledge and examples. We utilized OpenAI's `text-embedding-ada-002` API service for all embedding calculations across different methods and base models.

## A.2 MATHEMATICAL REASONING

**Benchmarks.** Our evaluation framework for assessing mathematical reasoning capabilities leverages two primary benchmarks: the GSM8K dataset, which comprises over 8,000 multi-step mathematical problems (Cobbe et al., 2021), and the GSM-HARD dataset, an adaptation of GSM8K where numbers in the questions are replaced with larger values to increase problem complexity (Gao et al., 2022). This study employs the PAL methodology to scrutinize the mathematical reasoning results, involving the utilization of Large Language Models (LLMs) to parse natural language problems, generate intermediary programmatic solutions, and subsequently execute these solutions via a Python interpreter. The test set for each benchmark consists of samples ranging from index 1 to 200. Uniquely, our approach does not use any examples for in-context learning, differing from the original PAL methods.

**Metrics and Baselines.** Accuracy serves as our principal metric for evaluation, aligning with the established metric for the GSM8K benchmark. Each question undergoes three execution attempts, with the average score recorded as the final result. The baselines, including DIRECT, CoT, RAG (1 shot), and RAG (5 shots), are consistent with those outlined in code generation, facilitating a comprehensive and comparative analysis across different code generation benchmarks. The RAG settings are consistent with the code generation tasks.

## A.3 EMBODIED PLANNING

We further conduct experiments on embodied planning benchmarks on open-ended environments Minecraft (Lin et al., 2023).

**Benchmarks.** The complexity and vast item interconnectivity within the open-world Minecraft present an ideal testbed for evaluating the LLM's capability to generate long-horizon plans (Yuan et al., 2023; Wang et al., 2023c;b). With thousands of items and intricate relationships between them, obtaining a specific item in survival mode from scratch may involve dozens of intermediate items and their quantitative relationships, such as crafting 1 crafting table from 4 planks. This setting rigorously tests the planning abilities of LLMs instead of low-level control policies (Cai et al., 2023b; Baker et al., 2022; Cai et al., 2023a; Lifshitz et al., 2023; Yuan et al., 2024). Moreover, Wang et al. (2023b) have identified instances of hallucinations about Minecraft knowledge in OpenAI's ChatGPT and a general scarcity of Minecraft-related knowledge in open-source language models, making this task a suitable benchmark for assessing the RAG algorithm's effectiveness.

The planning prompts are aligned with those used in DEPS (Wang et al., 2023c), structured as Python templates and evaluated using MC-TextWorld as detailed by Lin et al. (2023). A set of 100 tasks were randomly selected for the test set, ranging from simple objectives like obtaining a crafting table to more complex goals such as crafting an iron helmet and even challenging making an enchanting table. The task instruction is formulated as:

- Give you nothing in the inventory, generate a step-by-step plan for the task of obtaining a {placeholder:acacia_boat} in Minecraft survival mode, and describe the object Minecraft item and its number at every step. For every step, start with 'STEP' as start.

- Give you nothing in the inventory, generate a step-by-step plan for the task of obtaining a {placeholder:diamond_pickaxe} boat in Minecraft survival mode, and describe the object Minecraft item and its number at every step. For every step, start with 'STEP' as start.

There are over 100 tasks involving different Minecraft items.

**RAG Settings.** For the retrieval component of the RAG algorithm, we utilized the Minecraft Wiki[5] and DigMinecraft[6] websites as the information sources accessible to the LLMs. Data from these websites was cleaned and formatted into markdown text, then segmented into trunks not exceeding 2000 tokens

---

[5]https://minecraft.wiki/
[6]https://www.digminecraft.com/

each, with embedding calculations performed using OpenAI's `text-embedding-ada-002` API service.

**Evaluation Metrics.** Based on the methodology of Huang et al. (2022), our evaluation of open-ended, long-horizon planning in Minecraft focuses on both executability and plausibility. Executability primarily examines whether a plan can be carried out, including the accuracy of each step's preconditions and effects. The executability is automatically calculated using MC-TextWorld (Lin et al., 2023). However, executability only evaluates if an objective-level plan can be executed, without considering the specific details involved in executing individual objectives. For instance, crafting a wooden pickaxe requires placing a crafting table and arranging three planks and two sticks in a particular pattern, which are important details for human execution but not assessed by MC-TextWorld. Therefore, we complement our evaluation with human ratings to assess the plausibility of plans.

### A.4 CREATIVE WRITING

To further understand the potential of Retrieval-Augmented Generation (RAG) models in enhancing the creativity and relevance of generated content, we extend our investigation to open-ended text generation tasks within the realm of creative writing.

**Benchmarks.** The versatility of RaR was tested through a series of creative writing tasks, each chosen to highlight different aspects of open-ended text generation. These tasks include:

- Write a survey paper to summarize the `placeholder:Retrieval-augmented Generation` methods for Large Language Models.
- Describe of `placeholder:Jin-Yong`'s life.
- Summarize the `placeholder:American Civil War` according to the timeline.

For each task, three variants for `placeholder` were created to ensure a comprehensive evaluation of the model's performance across different contexts and requirements.

**RAG Settings.** Differing from previous tasks, creative writing is categorized as an open-ended generation task, demanding a broader scope of information retrieval to aid content generation. To accommodate this, Google was utilized as the search engine, with the top-k web pages converted into markdown text to assist the LLM in generating outputs. This approach allowed LLM to leverage a wide array of information sources.

**Baselines and Evaluations.** To benchmark RaR's performance, we compared it against DIRECT, RAG-1 shot, and RAG-5 shot methods, all based on the `gpt-3.5-turbo` model. The evaluation was conducted by human experts, employing the TrueSkill rating system (Herbrich et al., 2006) to calculate scores for each method. This evaluation framework enabled a comprehensive assessment of each model's creative output quality, accuracy, relevance, and innovativeness.

## B PROMPT DETAILS

Our prompts consist of three parts: prompt for generating initial answer, prompt for generating search query, and prompt for revising answers according to retrieved context.

---

**Prompt B.1: Prompt for generating initial answers in creative writing tasks**

{**user**}
##Question:
**{question}**
##Instruction:
Try to answer this question/instruction with step-by-step thoughts and make the answer more structural.
Use /n/n to split the answer into several paragraphs.
Just respond to the instruction directly. DO NOT add additional explanations or introducement in the answer unless you are asked to.
{**assistant**}
...

---

The process of query generation is omitted in code generation tasks. Instead, we use the generated code draft as a query and compute the embedding of it based on OpenAI Embedding services. For embodied planning and creative writing tasks, we will generate an additional query.

---

**Prompt B.2: Prompt for generating open-search query in creative writing tasks**

##Question:
**{question}**
##Content:
**{answer}**
##Instruction:
I want to verify the content correctness of the given question, especially the last sentences.
Please summarize the content with the corresponding question.
This summarization will be used as a query to search with Bing search engine.
The query should be short but need to be specific to promise Bing can find related knowledge or pages.
You can also use search syntax to make the query short and clear enough for the search engine to find relevant language data.
Try to make the query as relevant as possible to the last few sentences in the content.
**IMPORTANT**
Just output the query directly. DO NOT add additional explanations or introduecle in the answer unless you are asked to.
{**assistant**}
...

---

**Prompt B.3: Prompt for revising answer according to retrieved materials in creative writing tasks**

{**user**}
##Existing Text in Wiki Web:
**{content}**
##Question:
**{question}**
##Answer:
**{answer}**
##Instruction:
I want to revise the answer according to retrieved related text of the question in WIKI pages.
You need to check whether the answer is correct.
If you find some errors in the answer, revise the answer to make it better.
If you find some necessary details are ignored, add it to make the answer more plausible according to the related text.
If you find the answer is right and do not need to add more details, just output the original answer directly.
**IMPORTANT**
Try to keep the structure (multiple paragraphs with its subtitles) in the revised answer and make it more structural for understanding. Split the paragraphs with /n/n characters. Just output the revised answer directly.
DO NOT add additional explanations or annoucement in the revised answer unless you are asked to.
{**assistant**}
...

## C    TRUESKILL EVALUATION FRAMEWORK

Part of the tasks in "Embodied planning" and "creative writing" involve using humans for labeling. Human labelers have 4 choices: "A is better", "B is better", "Tie" or "Both are bad". In this case, "Tie" and "Both are bad" will be counted as a tie. For each task group, we have selected more than 10 professional annotators to provide labels. We use the Python "trueskill" package to calculate the win rate and score. The default score for every method is set as 25. In order to facilitate user understanding and selection, we also provide prompts when entering the system.

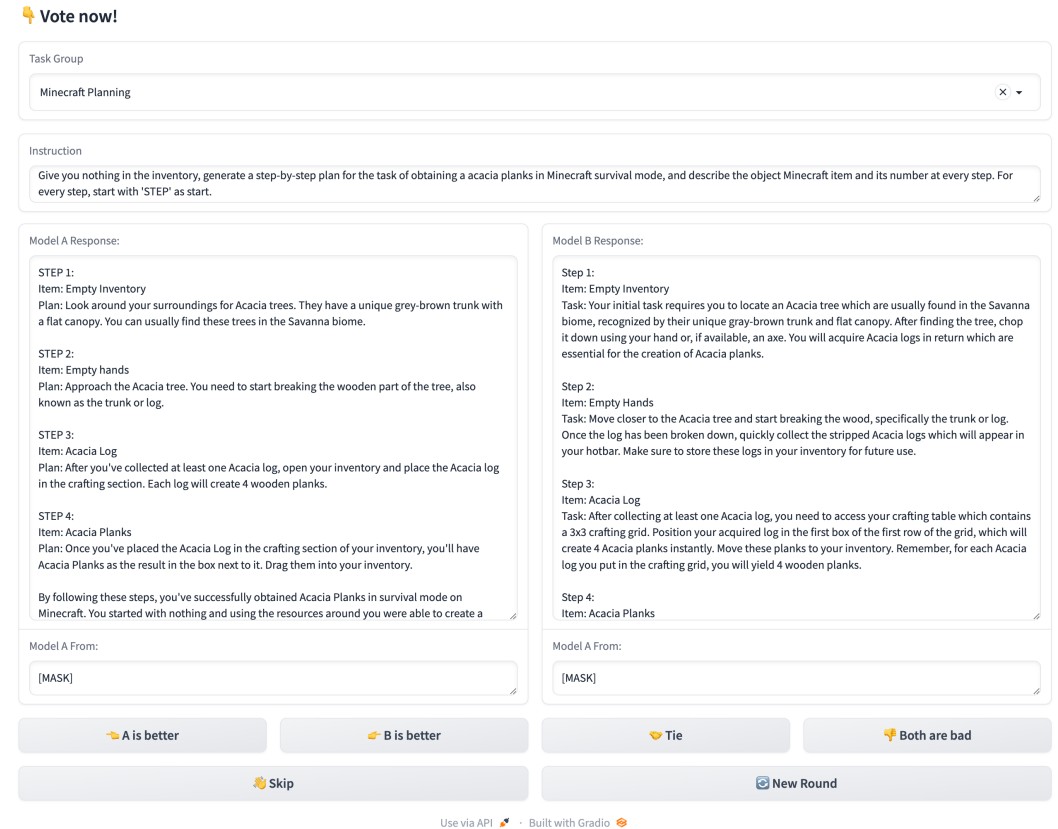

Figure C.1: The human evaluation UI. We will display responses from two different methods for the same instruction on the page simultaneously. The source of the response will be marked as [MASK], and after human labeling, [MASK] will be replaced with the specific method name.

# Chatbot Arena : Benchmarking LLMs in the Wild
##Rules
- Refresh to obtain the question and its corresponding answers from two anonymous models.
- Vote for the better answer. And then click "New Round" to get a new question.
- If both answers are bad, vote for "Both are bad".
- If you want to skip, click "Skip".

## Principle
You can evaluate the performance of the model from the following aspects:
1. **Relevance**: Does it answer the question accurately?
2. **Accuracy**: Is it accurate? For example, a crafting table is made by combining 4 wooden planks, not 4 logs; a diamond axe requires 3 diamonds and 2 sticks to craft, not 3 sticks and 2 diamonds.
3. **Completeness**: Is it complete? For example, crafting a wooden pickaxe from logs requires first crafting wooden planks and then crafting sticks before finally being able to craft the pickaxe. The intermediate steps cannot be ignored.
4. **Readability**: Is it coherent?
5. **Executability**: Considering the characteristics of the game, is it executable?

## Vote now!

# D    DISSCUSSIONS ON BENCHMARK CONTAMINATION

To avoid the code library containing solutions to the exact problems being evaluated) in code generation benchmarks, we adopted a rigorous pre-processing methodology as described by Guo et al. (2024). This process was meticulously designed to remove any direct matches or overly similar

code snippets from our search vector library, thereby ensuring that our evaluation remains fair and uncontaminated by pre-existing solutions. This examination aims to underscore the importance of maintaining the integrity of the evaluation process while utilizing external knowledge sources to augment the capabilities of language models in code-generation tasks.

Table 4: Benchmark contamination study based on CodeLlama-7b-Python models. We find using the code corpus finetuning the CodeLlama models directly will damage model performance on the code generation benchmark, while using RaR for in-context generation yields better performance.

| Method | HumanEval | | HumanEval+ | |
|---|---|---|---|---|
| | pass@1 | pass@5 | pass@1 | pass@5 |
| DIRECT | 40.85% | 53.65% | 37.43% | 48.78% |
| FINETUNE | 29.02% | 40.24% | 26.34% | 35.98% |
| RaR | 45.73% | 59.75% | 43.29% | 53.66% |

To further explore the potential benchmark contamination, we also conducted additional finetuning on CodeLLaMA-7B-Python using the code corpus in Table 4.

# E  MORE RESULTS

## E.1  EMBODDIED PLANNING

Embodied planning involves multiple steps, each of which relies on specific world knowledge and causal knowledge (i.e., preceding steps are usually preconditions for subsequent steps), including recipes for items in Minecraft, tools for performing related actions, and quantity restrictions. Therefore, although the plan generated by ChatGPT may appear complete and correct, there are often errors within the steps that can affect the proper execution of the plan.

We mark the errors found in the generated plan in red.

---

**Response E.1:** Response from Zero-Shot CoT(based on ChatGPT-3.5) in embodied planning tasks

{**instruction**}
Give you nothing in the inventory, generate a step-by-step plan for the task of obtaining a golden apple in Minecraft survival mode, and describe the object Minecraft item and its number at every step. For every step, start with 'STEP' as start.

{**response** from Original ChatGPT}
STEP 1: Start by punching trees to gather at least 4 logs. - Minecraft item: 4x Logs
STEP 2: Craft the logs into a crafting table. - Minecraft item: 1x Crafting Table
STEP 3: Place the crafting table on the ground and right-click on it to open the crafting interface.
STEP 4: Use the crafting interface to turn 4 of the oak planks into a wooden pickaxe.
- Minecraft item: Wooden Pickaxe
STEP 5: Find a cave or dig into the ground to gather some stone. Mine at least 8 blocks of stone using your wooden pickaxe. - Minecraft item: 8x Cobblestone
STEP 6: Craft the cobblestone into a furnace using the crafting table. - Minecraft item: 1x Furnace
STEP 7: Place the furnace on the ground and right-click on it to open the furnace interface.
STEP 8: Use the furnace interface to smelt some iron ore. Place at least 3 iron ore in the top slot of the furnace and use wooden planks as fuel in the bottom slot. - Minecraft item: 3x Iron Ingots
STEP 9: Craft the iron ingots into a bucket using the crafting table. - Minecraft item: 1x Bucket
STEP 10: Find a water source (such as a lake or a river) and right-click on it while holding the bucket to fill it with water. - Minecraft item: 1x Water Bucket
STEP 11: Look for an apple tree in a forest biome. Break some leaves until you find an apple. - Minecraft item: 1x Apple
STEP 12: Combine the water bucket and the apple in the crafting table to craft a golden apple. - Minecraft item: 1x Golden Apple
Congratulations! You have successfully obtained a golden apple in Minecraft survival mode.

---

Although the Zero-shot CoT has generated a step-by-step plan overall, there are many factual errors within it. These include recipe errors in STEP 2, where the crafting table requires planks instead of wood; missing raw materials in STEP 4, as the wooden pickaxe needs 2 sticks but lacks the relevant

step in the plan; absence of instructions to use a stone pickaxe to mine iron ore in STEP 8; and an incorrect recipe for golden apple in STEP 12, which should include gold ingots and an apple rather than a water bucket.

---

**Response E.2: Response from RAR(based on ChatGPT-3.5) in embodied planning tasks**

{**instruction**}
Give you nothing in the inventory, generate a step-by-step plan for the task of obtaining a golden apple in Minecraft survival mode, and describe the object Minecraft item and its number at every step. For every step, start with 'STEP' as start.

{**response** from Original ChatGPT}

STEP 1: Find a tree and chop it down to get wood. You need at least 4 Oak Logs.
- Minecraft items: 4x Oak Logs
STEP 2: Craft the oak log into oak planks.
- Minecraft items: 16x Oak Planks
STEP 3: Convert some oak planks into sticks.
- Minecraft items: 4x Stick
STEP 4: Use the crafting table to create a wooden pickaxe.
- Minecraft items: 1x Wooden Pickaxe
STEP 5: Start digging down to find stone and gather at least three pieces of cobblestone.
- Minecraft items: 3x Cobblestone
STEP 6: Use the crafting table to create a stone pickaxe.
- Minecraft items: 1x Stone Pickaxe
STEP 7: Explore caves and mineshafts to find iron ore. Collect 3 iron ore for crafting an iron pickaxe.
- Minecraft items: 3x Iron Ore
STEP 8: Smelt the iron ore using a furnace to obtain iron ingot.
- Minecraft items: 3x Iron Ingot
STEP 9: Use the crafting table to create an iron pickaxe.
- Minecraft items: 1x iron pickaxe
STEP 10: Explore caves and mineshafts to find gold ore. Use an iron pickaxe or higher to mine the gold ore block. Collect as much gold ore as needed for crafting a golden apple.
- Minecraft items: 8x Gold Ore
STEP 11: Smelt the raw gold using a furnace to obtain a gold ingot.
- Minecraft items: 8x Gold Ingot
STEP 12: Find an oak tree and chop it down until an apple falls.
- Minecraft items: 1x Apple
STEP 13: Open the crafting menu and place the items in the correct pattern to craft a Golden Apple. In the 3x3 crafting grid, place 8 Gold Ingots in the first and third rows, leaving the middle square empty. In the second row, place 1 Gold Ingot in the first and third squares, and place an Apple in the middle square.
- Minecraft items: 1x Golden Apple
Congratulations! You have successfully obtained a golden apple in Minecraft survival mode.

---

There are still errors in the plan generated by RAR, such as the lack of tools before "STEP 8: smelt iron ore into iron ingots", which should be "Mine 8 cobblestone to craft 1 furnace". However, compared to the errors in ChatGPT, the error rate in the plan has been significantly reduced.

We have also listed the links of the retrieved pages involved in different steps in Table 5 and Table 6. We can see that the text sources retrieved in each step generated by RaR are usually highly related to the synthesized item of that step. Traditional RAG (with 5 retrieval documents) uses instructions for retrieval and can only find the final step and other unrelated items, which may even harm the model's output. Compared to standard RAG, RaR extracts more pertinent knowledge from the database and provides more accurate answers. Furthermore, RaR does not directly generate answers based on retrieved content but evaluates previously generated content using both external information and internal knowledge. If inconsistencies or inaccuracies are detected in previous responses, revisions are made accordingly. By leveraging LLM's reasoning capabilities, RaR can decrease its reliance on retrieved data.

### E.2 CREATIVE WRITING

**Discussions on Computational Efficiency.**

Table 5: Page link of retrieved text by RaR in embodied Minecraft planning tasks.

| Step | Item | Recipe | Link |
|---|---|---|---|
| 1 | 4x Oak Log | - | https://minecraft.fandom.com/wiki/Log |
| 2 | 16x Oak Planks | 4x Oak Log | https://www.digminecraft.com/basic_recipes/make_oak_wood_plank.php |
| 3 | 4x Stick | 2x Oak Planks | https://www.digminecraft.com/basic_recipes/make_stick.php |
| 4 | 1x Wooden Pickaxe | 3x Oak Planks, 2 Stick | https://www.digminecraft.com/tool_recipes/make_wooden_pickaxe.php |
| 5 | 3x Cobblestone | Wooden Pickaxe | https://minecraft.fandom.com/wiki/Cobblestone |
| 6 | 1x Stone Pickaxe | 3x Cobblestone, 2 Stick | https://www.digminecraft.com/tool_recipes/make_stone_pickaxe.php |
| 7 | 3x Iron Ore | Stone Pickaxe | https://minecraft.fandom.com/wiki/Iron_Ore |
| 8 | 3x Iron Ingot | 3x Iron Ore | https://www.digminecraft.com/basic_recipes/make_iron_ingot.php |
| 9 | 1 Iron Pickaxe | 3x Iron Ingot, 2x Stick | https://www.digminecraft.com/tool_recipes/make_iron_pickaxe.php |
| 10 | 8x Gold Ore | Iron Pickaxe | https://minecraft.fandom.com/wiki/Gold_Ore |
| 11 | 8x Gold Ingot | 8x Gold Ore | https://www.digminecraft.com/basic_recipes/make_gold_ingot.php |
| 12 | 1x Apple | - | https://minecraft.fandom.com/wiki/Apple |
| 13 | 1x Golden Apple | 8x Gold Ingot, 1x Apple | https://www.digminecraft.com/food_recipes/make_golden_apple.php |

Table 6: Page link of retrieved text by conventional RAG methods in embodied Minecraft planning tasks.

| Step | Item | Recipe | Link |
|---|---|---|---|
| 1 | 4x Oak Log | - | - |
| 2 | 16x Oak Planks | 4x Oak Log | - |
| 3 | 4x Stick | 2x Oak Planks | - |
| 4 | 1x Wooden Pickaxe | 3x Oak Planks, 2 Stick | - |
| 5 | 3x Cobblestone | Wooden Pickaxe | - |
| 6 | 1x Stone Pickaxe | 3x Cobblestone, 2 Stick | - |
| 7 | 3x Iron Ore | Stone Pickaxe | - |
| 8 | 3x Iron Ingot | 3x Iron Ore | - |
| 9 | 1 Iron Pickaxe | 3x Iron Ingot, 2x Stick | - |
| 10 | 8x Gold Ore | Iron Pickaxe | - |
| 11 | 8x Gold Ingot | 8x Gold Ore | - |
| 12 | 1x Apple | - | https://minecraft.fandom.com/wiki/Apple |
| 13 | 1x Golden Apple | 8x Gold Ingot, 1x Apple | https://minecraft.fandom.com/wiki/Golden_Apple
https://www.digminecraft.com/food_recipes/make_golden_apple.php
https://minecraft.fandom.com/wiki/Enchanted_Golden_Apple |
| 14 | Other unrelated Items | - | https://www.digminecraft.com/game_commands/enchant_command.php
https://www.digminecraft.com/tool_recipes/make_golden_pickaxe.php |

In our experiment, we used RAG-5 as the baseline, which retrieves 5 content sources (commonly the default choice for most current RAG algorithms). We also set an upper limit on RaR retrieval times, ensuring that RaR does not retrieve more than 5 contents. Therefore, RaR does not exceed the standard RAG algorithm in terms of content retrieval.

Compared to including all 5 retrieved contents at once in context for generation in RAG-5, RaR incorporates one external content at a time to reduce prompt length. This approach prevents exceeding the language model's context length limitation that may occur when inputting all knowledge into the prompt with RAG-5. Using only one external content per revision in RaR avoids issues related to excessive content length and allows more token space for generating longer content.

Additionally, we utilized the Google search engine directly for RaR searches and compared time consumption between RaR and RAG-5 when creating creative writing prompts. Our findings indicate that while increased search time plays a role, more time is consumed during crawling and generation processes. And RaR does not have a significant efficiency gap compared to RAG-5. The detailed time consumption of every method is listed in Table 7. However, since RaR must be processed causally, it is worth further exploring how to improve the efficiency of RaR by referring to speculative decoding and other methods (Leviathan et al., 2022).

Table 7: Comparison of time consumed by different methods in creative writing tasks.

| Method | CoT | RAG-5 | RaR |
|---|---|---|---|
| Search Time(s) | - | 0.12 | 0.47 |
| Crawl Content Time(s) | - | 5.17 | 4.26 |
| Generation Time(s) | 3.59 | 10.72 | 12.75 |
| Total Time(s) | 3.59 | 17.37 | 18.45 |

---

**Response E.3: Response from ORIGINAL ChatGPT-3.5 in creative writing tasks**

{**instruction**}
Describe the history of the United States. {**response** from Original ChatGPT}

The history of the United States dates back thousands of years, with indigenous peoples such as the Native Americans inhabiting the land long before the arrival of European settlers. However, the official history of the United States begins in 1776 when the Founding Fathers declared independence from Great Britain. This marked the start of the American Revolution, a conflict that ultimately led to the creation of the United States as an independent nation.
After gaining independence, the United States went through a period of rapid expansion in the 19th century. The country acquired vast amounts of territory, including the Louisiana Purchase and the annexation of Texas. This expansion was accompanied by conflicts, such as the Mexican-American War and the Civil War, which tested the unity of the nation.
In the 20th century, the United States emerged as a global superpower. The country played a significant role in both World War I and World War II, eventually becoming the leader of the free world during the Cold War with the Soviet Union. The United States experienced social and cultural shifts during this time, including the Civil Rights Movement and the rise of consumerism.
Today, the United States remains a powerful nation, with a diverse population and a complex history that continues to shape its identity.

---

**Response E.4: Response from Zero-shot CoT in creative writing tasks**

{**instruction**}
Describe the history of the United States.
{**response** from Zero-shot CoT}
**Pre-Columbian Era**
The history of the United States begins long before the arrival of European explorers in the 15th century. The land was initially inhabited by diverse Native American tribes, such as the Apache, Sioux, Iroquois, and Cherokee. These societies had rich cultures, with their own languages, traditions, and social structures. They lived off the land, flourishing through hunting, fishing, farming, and trade. However, the indigenous populations faced significant challenges, including conflicts, displacement, and diseases brought by the European settlers.
**Colonial Period**
In the 16th century, European powers began to claim territories in what is now the United States. The Spanish were the first to establish colonies, initially focusing on Central and South America. Later, English, French, and Dutch explorers arrived, establishing their own colonies along the Atlantic coast. These colonies varied in purpose and organization, with the English colonies primarily seeking economic opportunities and religious freedom. Over time, thirteen British colonies emerged, forging their own identities, developing local governments, and experiencing an increase in population due to immigration.
**American Revolution and Beyond**
Tensions between the British colonies and their colonial masters grew, leading to the American Revolution in 1775. The colonies, unified under the Continental Congress, fought for independence from British rule. The war lasted several years, until the signing of the Treaty of Paris in 1783, which granted the newly formed United States of America independence. After achieving independence, the founding fathers convened to draft the Constitution, establishing the framework for the new nation's government.
This is just a brief overview, and the history of the United States is vast and complex. However, these steps provide a simplified understanding of how the nation evolved from Native American civilizations to a group of British colonies, to an independent country.

**Response E.5: Response from RAG(1-shot) in creative writing tasks**

{**instruction**}
Describe the history of the United States.
{**response** from RAG-1}
The history of the United States is a complex and rich tapestry that spans over four centuries. The story begins with the arrival of European explorers and settlers in the 16th century. In 1492, Christopher Columbus reached the Americas, which led to subsequent expeditions and colonization by the Spanish, French, Dutch, and English. The first successful English settlement was established at Jamestown, Virginia in 1607.
Over the next few centuries, the colonies grew and developed their own unique cultures, economies, and systems of government. Tensions rose between the American colonies and the British Crown, leading to the American Revolutionary War in 1775. The colonies declared their independence in 1776 and formed the United States of America, which became a republic based on democratic ideals and principles.
The young nation faced numerous challenges in its early years, including establishing a central government, expanding westward, and dealing with the issue of slavery. The 19th century saw rapid westward expansion, the Industrial Revolution, and the Civil War. The United States emerged from the Civil War as a more unified and powerful nation, with the abolition of slavery and the preservation of the Union. In the 20th century, the United States became a global superpower, playing a crucial role in both World Wars and the Cold War. It witnessed significant social movements for civil rights and equality, as well as advancements in technology and innovation.
Today, the United States is a diverse and influential country with a strong economy, global military presence, and a complex political landscape. Its history continues to shape its present and future, as the nation grapples with ongoing social, economic, and political challenges.

**Response E.6: Response from RAG(5-shot) in creative writing tasks**

{**instruction**}
Describe the history of the United States.
{**response** from RAG-5}
The history of the United States dates back to the colonization of the Americas by European explorers in the 16th century. The first permanent English settlement was established in Jamestown, Virginia in 1607. This marked the beginning of British colonization in North America.
Over the next century, more British colonies were established along the eastern seaboard, including Massachusetts, New York, Pennsylvania, and Georgia. These colonies grew and developed their own distinct cultures and economies.
Tensions between the colonies and the British government grew in the mid-18th century, leading to the American Revolutionary War in 1775. The colonists, led by figures such as George Washington, Thomas Jefferson, and Benjamin Franklin, fought for their independence from British rule.
In 1776, the Second Continental Congress adopted the Declaration of Independence, which declared the thirteen American colonies to be independent states. This marked the birth of the United States of America.
Following the Revolutionary War, the Founding Fathers drafted the Constitution of the United States, which established a federal government and outlined the rights and freedoms of American citizens. The Constitution was ratified in 1788 and is still the supreme law of the land.
Since its founding, the United States has experienced significant growth and development. It expanded westward through the Louisiana Purchase and the annexation of Texas, and it eventually stretched from coast to coast. The country also faced significant challenges, such as the Civil War, the Great Depression, and the Civil Rights Movement.
Today, the United States is a global superpower with a diverse population and a strong economy. It is known for its democratic system of government, individual freedoms, and cultural influence.

> **Response E.7: Response from RaR in creative writing tasks**
>
> {**instruction**}
> Describe the history of the United States.
> {**response** from RAR}
> The history of the United States is characterized by several significant phases. The pre-colonial period witnessed diverse Native American populations with unique cultures inhabiting the land. This era transitioned into colonialism through European exploration and settlement by powers such as the Spanish, French, Dutch, and British, leading to the establishment of colonies along the East Coast with distinct economic and political frameworks.
>
> A pivotal moment in American history was the American Revolution, beginning in 1775, as the colonies sought independence from British rule. The Declaration of Independence in 1776 marked the nation's founding on principles of liberty and democracy, establishing autonomy from Great Britain. This revolutionary period not only birthed a republic based on the will of the people but also forged American national identity, fostering a collective ethos entrenched in history and culture, inspiring revolutions worldwide and aiding in the establishment of republican governments elsewhere.
>
> The United States stands out as the first modern nation to achieve independence through a national war of liberation, articulating its motives and aspirations in a declaration of independence. This event had global reverberations, spreading fundamental values like liberty, equality, natural and civil rights, and civic duty. The formative years were shaped by the drafting of the Constitution in 1787, defining the federal government's framework. The subsequent ratification of the Bill of Rights further cemented individual freedoms and outlined governance principles that have endured across the nation's history, ensuring a system of checks and balances that protect citizens' rights.
>
> The 19th century in the United States witnessed significant developments, including westward expansion, industrialization, and conflicts like the Civil War (1861-1865), which decisively addressed issues of slavery and reinforced federal authority over the states. The late 19th and early 20th centuries were marked by rapid industrial growth, extensive immigration, urbanization, and overseas expansion, including involvement in conflicts such as the Spanish-American War and World War I. The 20th century saw the United States emerge as a global superpower after World War II, engaging in the Cold War with the Soviet Union and participating in conflicts like the Korean and Vietnam Wars, significantly impacting the nation's foreign policy and global standing.
>
> Recent history includes social movements for civil rights, gender equality, and environmental protection, as well as economic shifts, advancements in technology, and challenges in areas such as healthcare and international relations.

