# OpenReview forum: "Optimizing Inference-Time Reasoning in LLMs via Retrieval-Augmented Reflection"
_ICLR.cc/2025/Conference — Submitted to ICLR 2025_

### Official Review · Reviewer_UX3s · 2024-10-16

**Soundness:** 1
**Presentation:** 3
**Contribution:** 3
**Rating:** 5
**Confidence:** 4

**Summary:**

This paper proposes RaR (Retrieval-augmented Reflection), which introduces retrieval-augmented reflection in multi-step reasoning, combining relevant information from external retrieval to correct the intermediate reasoning steps, thereby improving the model's reasoning performance.

Specifically, the proposed method first uses Zeroshot CoT to generate a step-by-step reasoning trajectory, in which may contain erroneous parts.
Subsequently, the paper attempts to improve the reasoning by retrieving relevant information from an external knowledge base through a retrieval-augmented approach, and using this information to correct any potential errors in the reasoning.
Chain-of-thought reasoning is step-by-step, and to achieve more fine-grained and precise corrections, this paper proposes to correct the reasoning steps one by one from the beginning to the end.
At each step of reasoning verification, relevant external information needs to be retrieved. Assuming that the current step i needs to be corrected, the question and reasoning steps 1 through i are used as the retrieval query to retrieve relevant materials, and the retrieved content is used to correct the reasoning.
The paper conducted experiments on tasks such as code generation, mathematical reasoning, and knowledge-intensive question answering.

**Strengths:**

1. The paper proposes a reasoning framework that combines retrieval-augmented and reflection, called RaR (Retrieval-augmented Reflection), which can correct potential errors in the reasoning process based on external knowledge.
2. The paper proposes a multi-round reflection method, where each round only retrieves and reflects on the current step, thereby more accurately identifying the location of errors. On another dimension, this approach can increase computational FLOPs during reasoning, allowing for inference-time scaling based on the inference scaling laws.
3. The paper conducted experiments on several datasets, including tasks such as code generation, mathematical reasoning, and question answering, demonstrating the effectiveness of the method.

**Weaknesses:**

1. The writing of method part in the paper (Line 216~Line 239) is somewhat vague. While I can understand the general idea of the proposed method, I cannot accurately grasp the specific details through the formulas and descriptions, and clarification from the authors is needed.
2. The author claims that by expanding n (the number of reasoning steps) to achieve Inference Scaling. However, this dimension seems not easy to expand. On one hand, the number of reasoning steps generated by Zeroshot CoT is not entirely controllable.  On the other hand, for a specific problem, the number of reasoning steps is clearly limited. These two aspects make it less feasible and scalable to increase the number of reasoning steps in order to increase the number of verification rounds.
3. The QA dataset used in this paper is TriviaQA, which only requires single-hop retrieval and reasoning to complete. Given the author's motivation to use retrieval to improve multi-step reasoning, using multi-hop reasoning datasets such as MuSiQue, 2WikiMHQA, etc., seems to be a more appropriate choice.

**Questions:**

1. In the paragraph from Line 220 to Line 224, the description and the formula on Line 224 seem to be incorrect or unclear. It is difficult to understand what the formula is intended to express.
2. What are the differences between the two sets of formulas on Line 219, 224, and Line 233~235?
3. The mathematical reasoning in the article uses simple arithmetic reasoning, with the GSM8K and GSM-Hard datasets. Unlike MATH, which involves some advanced mathematical knowledge, these two datasets consist of simple multi-step arithmetic operations and basic linear equations, without involving external knowledge. Secondly, the retrieval for mathematical reasoning uses Jupyter code corpus, which does not seem to be helpful for solving mathematical reasoning problems.

---

> ### Author Response · Authors · 2024-11-23
>
> > The writing of method part in the paper (Line 216~Line 239) is somewhat vague. While I can understand the general idea of the proposed method, I cannot accurately grasp the specific details through the formulas and descriptions, and clarification from the authors is needed.
>
>
> Thank you for your reply. We have rewritten the method section of RaR based on your feedback. In Section 3.1, we first introduce how the standard Retrieval-Augmented Reflection is implemented and the differences between RaR and Retrieval-augmented Generation (RAG). In Section 3.2, we demonstrate how Iterative RaR iteratively updates intermediate reasoning steps and the final response. In Section 3.3, we show how RaR and other baseline methods scale when given more inference-time tokens. We have also updated Algorithm 1 to make it easier for readers to understand the RaR pipeline.
>
> If you have any further questions about the method, feel free to discuss them with me.
>
> > The author claims that by expanding n (the number of reasoning steps) to achieve Inference Scaling. However, this dimension seems not easy to expand. On one hand, the number of reasoning steps generated by Zeroshot CoT is not entirely controllable. On the other hand, for a specific problem, the number of reasoning steps is clearly limited. These two aspects make it less feasible and scalable to increase the number of reasoning steps in order to increase the number of verification rounds.
>
> Thank you for your reply. Iterative RaR does not explicitly increase the number of CoT steps to expand n. During the reflect and revise process, Iterative RaR focuses on: 1. the accuracy of reasoning steps and 2. the consistency of the final response. When generating retrieval queries, we let RaR first focus on the reasoning steps, usually m steps correspond to m times of RaR. Finally, when m times of RaR still do not reach the maximum token limitation, we will further perform RaR on the complete results to continue scaling. The specific RaR process can be referenced in our updated Algorithm 1 and Equations (5) and (6), which detail the formulation during the RaR scaling process.
>
> Therefore, RaR does not need to control the generation steps of CoT. Sorry for the confusion, we have revised the paper according to your comments.
>
>
> > The QA dataset used in this paper is TriviaQA, which only requires single-hop retrieval and reasoning to complete. Given the author's motivation to use retrieval to improve multi-step reasoning, using multi-hop reasoning datasets such as MuSiQue, 2WikiMHQA, etc., seems to be a more appropriate choice.
>
> Thank you for your comment. Based on your feedback, we have added the 2WikiMultiHopQA and Musique benchmark. All methods use `gpt-4o-mini` and are tested under a maximum of 4K tokens.
>
> | Method | DIRECT | RAG | Active RAG | IRCoT | RaR(4K) | RaR(8K) |
> | --- | --- | --- | --- | --- | --- | --- |
> | 2WikiMultiHopQA Answer F1 | 50.3 | 63.7 | 67.9 | 68.4 | 77.6 | 79.4 |
> | Musique Answer F1 | 41.9 | 44.6 | - | 56.5 | 63.4 | - |
>
> Due to time constraints, we have currently only completed these baselines (RAG, Active RAG, IRCoT) and RaR (under 4K maximum tokens) and RaR (under 8K maximum tokens). Under the same token limitation, iterative RaR has shown better performance compared to other baselines, with a +21.8% relative improvement on Answer F1 score on 2WikiMultihopQA compared to RAG. When RaR is under more tokens (8k), the performance further improves (79.4 vs. 77.6 on Answer F1 score).
>
> We will add this experiment to the main text of the paper once all the baselines are completed.

---

> ### Author Response · Authors · 2024-11-23
>
> > 1. In the paragraph from Line 220 to Line 224, the description and the formula on Line 224 seem to be incorrect or unclear. It is difficult to understand what the formula is intended to express.
>
> Sorry for the confusion.
>
> This formula is to illustrate how RaR iteratively updates reasoning steps. To make the expression clearer, we have updated section 3.2 of the method. The updated formula is:
>
> $y_{i}^{\text{RaR}} \sim p_\text{LM}(\cdot \mid x, y_{i}^{\text{thought}}, V_i^k, y_i^\text{reflection}), i = 1$
>
> $y_{i}^{\text{RaR}} \sim p_\text{LM}(\cdot \mid x, y^{\text{RaR}}_{i-1}, y_i^{\text{thought}}, V_i^k, y_i^\text{reflection}),1 < i < J,$
>
> $y_{i}^{\text{RaR}} \sim p_\text{LM}(\cdot \mid x, y_{i-1}^{\text{RaR}}, y_{i}^{\text{thought}}, y^{\text{raw}}, V_i^k, y_i^\text{reflection}), i = J. $
>
> In this equation, $y^{RaR}_i$ represents the RaR response during the i-th iteration process, $y_i^{thought}$ represents the i-th step of reasoning, $V_i^k$ represents the document most relevant to the current i-th step, and $y_i^{reflection}$ represents the reflection of the LLM based on the content of the retrieval for the i-th step.
>
> We have updated the Method 3.2 Section and Algorithm 1 part of the paper based on your feedback.
>
> > 2. What are the differences between the two sets of formulas on Line 219, 224, and Line 233~235?
>
> Sorry for the confusion. Line 219 and 224 show the formulation of the query and RaR response, while in Line 233-235 we demonstrate how iterative RaR performs retrieval and reflection.
>
> Based on your feedback, we have updated these formulas in Section 3.2 of the paper. In the updated paper, the query for the i-th iteration of iterative RaR is shown in Equation (4):
>
> $q^i \sim p_{\text{LM}}(\cdot \mid x^*, \\{y_j^\text{thought}\\}_{j=1}^{j<=i}), i=1,\ldots,J.$
>
> Among them, J represents the number of reasoning steps, $x^*$ represents the instruction, and $y_j^{thought}$ represents the content of the j-th reasoning step.
>
> The reflected response of iterative RaR in the i-th iteration (Equation (5)) is:
>
> $y_{i}^{\text{RaR}} \sim p_\text{LM}(\cdot \mid x, y_{i}^{\text{thought}}, V_i^k, y_i^\text{reflection}), i = 1$
>
> $y_{i}^{\text{RaR}} \sim p_\text{LM}(\cdot \mid x, y^{\text{RaR}}_{i-1}, y_i^{\text{thought}}, V_i^k, y_i^\text{reflection}),1 < i < J,$
>
> $y_{i}^{\text{RaR}} \sim p_\text{LM}(\cdot \mid x, y_{i-1}^{\text{RaR}}, y_{i}^{\text{thought}}, y^{\text{raw}}, V_i^k, y_i^\text{reflection}), i = J. $
>
> In this equation, $y^{RaR}_i$ represents the RaR response during the i-th iteration process, $y_i^{thought}$ represents the i-th step of reasoning, $V_i^k$ represents the document most relevant to the current i-th step, and $y_i^{reflection}$ represents the reflection of the LLM based on the content of the retrieval for the i-th step.
> Thanks for your comments.
>
>
> > 3. The mathematical reasoning in the article uses simple arithmetic reasoning, with the GSM8K and GSM-Hard datasets. Unlike MATH, which involves some advanced mathematical knowledge, these two datasets consist of simple multi-step arithmetic operations and basic linear equations, without involving external knowledge. Secondly, the retrieval for mathematical reasoning uses Jupyter code corpus, which does not seem to be helpful for solving mathematical reasoning problems.
>
> Thank you for your comments. Based on your feedback, we have added the experimental results of RaR on the MATH dataset. In the MATH benchmark, we uniformly used the Deepseek-math-7B model and used the PRM800k dataset as the retrieval documents library, with a maximum token limitation of 4K tokens.
>
> The experimental results are as follows:
>
> | Method | DIRECT | RAG | IRCoT | RaR |
> | --- | --- | --- | --- | --- |
> | Accuracy | 58.8 | 60.2 | 64.5 | 69.3 |
>
> Under the same model and RAG settings, RaR demonstrated a +15.1% relative improvement. We will further increase more experiments to demonstrate the robustness and scalability of RaR.

---

### Official Review · Reviewer_Pozp · 2024-10-28

**Soundness:** 3
**Presentation:** 2
**Contribution:** 3
**Rating:** 6
**Confidence:** 4

**Summary:**

This paper uses LLMs to generate chains-of-thought (CoT) for coding, math, and planning tasks, then uses RAG to iteratively improve each individual step in the original CoT multiple times to improve performance at the cost of extra compute and time during inference.  Their method outperforms many other baselines (including RAG, CoT, Self-Refine, and others) and also shows performance increases as models scale in parameters and more iterations of refinements are done. Finally, they show that refining a CoT with a single prompt in one generation is not as powerful as their method, where you fix each individual step of the CoT one by one.

**Strengths:**

1. Strong results across many commonly used baselines showing their method (RaR) outperforms all of them.
2. Timely and well-motivated, I think scaling during inference is probably the easiest way to get better outputs from closed-source LLMs right now.

**Weaknesses:**

1. Lack of an important baseline, RaG + CoT. There already are a ton of baselines here, but I do feel that RAG + CoT is a very close baseline to the method proposed and is a very common one that I think people would want to see before using RaR.  I think this baseline would also highlight (similarly to CoT+RAG in Table 3) that even with all the documents/passages required to answer a question, the LMs still benefit from iterative refinement.  This would be a big win for the paper and make it extremely clear to other researchers why RaR should be used.

2. More thorough analysis of why the baselines are failing.  Tables 3 and 4 are good ablations showing the need for iteratively refining each individual step without showing the full CoT, but I think the paper would also benefit from detailing where traditional methods are failing.  For example, if we used CoT+RAG or RAG+CoT with all the same documents retrieved in RAR, would we get similar performances?  I think more discussion on where RaR is outperforming the other baselines would help researchers understand why RaR is an effective method and where it can be improved for future work.

**Questions:**

- Have you tried matching the amount of compute for RAR with other baselines like Self-refine?
- Have you tried a baseline similar to RAR but without RAG? (Just asking the LLM to verify/fix steps with no additional context to establish how important the retrieved documents are)
- "With lower inference-time computation (FLOPs), a small LM can surpass the performance of the LM with more than 10 times parameters." I wasn't sure what this meant in the abstract.  Are you saying a small LM can outperform a larger one when there is a large compute budget and RAR is used?

---

### Official Review · Reviewer_yr3i · 2024-10-30

**Soundness:** 2
**Presentation:** 3
**Contribution:** 2
**Rating:** 3
**Confidence:** 3

**Summary:**

Many research efforts focus on correcting misinformation generated by language models through retrieval augmentation. This paper explores how iteratively revising a chain of thoughts with the help of information retrieval significantly improves large language models' reasoning abilities across various tasks, such as code generation, mathematical reasoning, and embodied planning. Experimental results show that, compared to methods like language model self-consistency and simple retrieval augmentation, the proposed RaR framework is more effective in mitigating hallucinations in the content generated by models.

**Strengths:**

This paper explores iteratively revising a chain of thoughts with the help of information retrieval to improve large language models' reasoning abilities. Its strengths include:

1. The writing in the paper is clear, and the figures are intuitive, effectively conveying the main ideas and supporting the overall arguments.

2. The method proposed by this paper incorporates a recursive correction mechanism by revising each thought step based on previously refined steps, allowing for continual consultation of relevant information. This significantly improves the accuracy and reliability of generated outputs compared to traditional RAG methods.

3. RaR is flexible in handling various tasks, such as code generation, mathematical reasoning, and embodied planning.

**Weaknesses:**

1. The baseline methods compared in Table1 are weak, and the proposed approach has not been compared with the latest related RAG research, which does not demonstrate the significant differences between RaR and existing work (Active-RAG, IRCoT, ...).

**Questions:**

1. Is the number of modification iterations in RaR related to the number of sentences in the answers? In that case, when comparing with RAG, is the number of retrievable instances in RAG consistent with that of RaR?

---

> ### Comment · Reviewer_yr3i · 2024-11-28
>
> Thank you for your feedback. After considering the insights shared by other reviewers, I have decided to retain the current rating score.

---

### Official Review · Reviewer_XHx9 · 2024-11-03

**Soundness:** 1
**Presentation:** 2
**Contribution:** 2
**Rating:** 3
**Confidence:** 4

**Summary:**

This paper studies the problem of leveraging RaG to improve LLMs’ reasoning capabilities, and proposes a new approach called Retrieval-augmented Reflection (RaR). RaR first generates an initial solution, and then alternates the process retrieval and revision to revise each step in the initial solutions. The paper evaluates the proposed approach on three domains and compares against a series of zero-shot baselines. The results suggest that the method can improve LLM performance on code generation and embodied planning tasks.

**Strengths:**

The results presented in the paper suggest good empirical performance.

The paper evaluates the method on multiple datasets spanning several domains.

**Weaknesses:**

I have several concerns, mainly regarding the evaluation. The comparison is probably unfair, which overstates the performance gains. Additionally, insufficient detail about the evaluation setup makes it difficult to validate the results. Specifically:

First, the comparison is unfair. The paper compares RaR against direct/CoT/self-consistency/RAG approaches. Among these, Direct/Cot and RAG are approaches that produce answers with one LLM call. In contrast, RaR, involves revising each reasoning step. IIUC, RaR would involve T LLM calls for producing an answer, where T is the number of steps, plus the additional costs of processing retrieved documents. To ensure fair comparison and to fully understand the differences of different approaches in terms of computation cost, I think it is necessary to 1) list out the computation cost needed for each approach (in terms of number of tokens) 2) compare methods under the same computation constraints.

Second, the missing details on the experimental setup makes it harder to interpret these results. Below are some concrete points:
* Table 1: Are all methods using the same computation budget? For self-consistency, how many samples are considered? For self-refine, how many rounds of evaluation are made?
* Table 2 (math reasoning and task planning sections): Which models are employed?
* Table 2 (QA section): The table structure is unclear. Different rows appear to use different base models (e.g., Reasoning row vs. RAG row), and the base models for the fourth RAG row are unspecified.

In particular, given the lack of description of experimental details. I also find some numbers seemingly need further verification.
In Table 1, GPT-4's reported direct performance on HumanEval and MBPP (57.3% and 60.0%) differs from the GPT-4 technical report (67.6% and 68.3% for zero-shot, pass@1).
* For instance, in Table 1, it is reported GPT-4 gets direct performance of 57.3% and 60.0 on human-eval and MBPP, respectively. But some other work, including GPT-4 tech report, and AgentCoder (Huang et al., 24), reports that GPT-4 gets a performance of 67.6 and 68.3 (zero-shot, pass@one) on HumanEval and MBPP respectively. It is unclear what leads to such mismatch.
* Table 2's math reasoning results show CoT significantly underperforming DIRECT, which seems counterintuitive.

The paper needs to provide comprehensive experimental details and restructure Tables 1 and 2 for clarity and accuracy.

Additionally, the paper lacks sufficient analysis of its results, especially for the qualitative outcomes on mathematical tasks. According to the results, the approach shows significant performance improvements in code generation and mathematical reasoning. But the underlying reasons for these improvements are not sufficiently explained. For code-related tasks, the improvements are somewhat intuitive as there often exist multiple code implementations with similar semantics.  For mathematical reasoning tasks, which are typically more problem-specific, it would be valuable to understand what drives such improvements. The paper does not provide analysis in the main body nor in the appendices to explain these performance gains. It would be good to provide some analysis around this.

**Questions:**

See weakness

---

### Meta-Review · Area_Chair_u2dg · 2024-12-23

**Metareview:**

The paper introduces Retrieval-Augmented Reflection (RaR), a framework for improving reasoning in LLMs by iteratively revising reasoning steps using external retrieval. While the method demonstrates potential in enhancing code generation, mathematical reasoning, and task planning, the submission contains substantial weaknesses. The evaluation lacks strong baselines and comprehensive comparisons, raising concerns about fairness. Details about experimental settings and results are insufficient, with discrepancies in reported performances. Methodological clarity is also lacking, particularly in the description of key components like stepwise reflection and scaling with inference-time computation.

Strengths include the focus on improving inference-time reasoning and promising empirical results. However, these are undermined by poor experimental rigor, weak baselines, and limited novelty.

**Additional Comments On Reviewer Discussion:**

Reviewers highlighted concerns about evaluation fairness, incomplete baselines, and unclear experimental settings. While the authors attempted to address these through additional experiments and clarifications, key issues remain unresolved. The lack of transparency and discrepancies in performance reporting led reviewers to question the validity of the results. Despite some improvements in clarity and methodology, the submission’s limitations justify a rejection, with encouragement for substantial revisions.

---

### Decision · Program_Chairs · 2025-01-22

Reject